# Comprehensive Integrated Analysis Reveals the Spatiotemporal Microevolution of Cancer Cells in Patients with Bone-Metastatic Prostate Cancer

**DOI:** 10.3390/biomedicines13040909

**Published:** 2025-04-09

**Authors:** Yinghua Feng, Xiuli Zhang, Guangpeng Wang, Feiya Yang, Ruifang Li, Lu Yin, Dong Chen, Wenkuan Wang, Mingshuai Wang, Zhiyuan Hu, Yuan Sh, Nianzeng Xing

**Affiliations:** 1Department of Epidemiology, School of Public Health, Shanxi Medical University, Taiyuan 030001, China; yinghua4172@163.com; 2Department of Urology, Shanxi Province Cancer Hospital/Shanxi Hospital Affiliated to Cancer Hospital, Chinese Academy of Medical Sciences/Cancer Hospital Affiliated to Shanxi Medical University, Taiyuan 030013, China; 3Department of Urology, National Cancer Center/National Clinical Research Center for Cancer/Cancer Hospital, Chinese Academy of Medical Sciences and Peking Union Medical College, Beijing 100021, China; 4State Key Laboratory of Molecular Oncology, National Cancer Center/National Clinical Research Center for Cancer/Cancer Hospital, Chinese Academy of Medical Sciences and Peking Union Medical College, Beijing 100021, China; 5Department of Rheumatology and Clinical Immunology, Peking University First Hospital, Beijing 100034, China; zhangxl2020@nanoctr.cn; 6CAS Key Laboratory for Biomedical Effects of Nanomaterials and Nanosafety, CAS Key Laboratory of Standardization and Measurement for Nanotechnology, CAS Center for Excellence in Nanoscience, National Center for Nanoscience and Technology, Beijing 100190, China

**Keywords:** prostate cancer, single-cell RNA sequencing, bone metastasis, metabolism and cytokines, spatiotemporal microevolution

## Abstract

**Background/Objectives:** Bone metastasis is a frequent and life-threatening event in advanced cancers, affecting up to 70–85% of prostate cancer patients. Understanding the cellular and molecular mechanisms underlying bone metastasis is essential for developing targeted therapies. This study aimed to systematically characterize the heterogeneity and microenvironmental adaptation of prostate cancer bone metastases using single-cell transcriptomics. **Methods:** We integrated the largest single-cell transcriptome dataset to date, encompassing 124 samples from primary prostate tumors, various bone metastatic sites, and non-malignant tissues (e.g., benign prostatic hyperplasia, normal bone marrow). After quality control, 602,497 high-quality single-cell transcriptomes were analyzed. We employed unsupervised clustering, gene expression profiling, mutation analysis, and metabolic pathway reconstruction to characterize cancer cell subtypes and tumor microenvironmental remodeling. **Results:** Cancer epithelial cells dominated the tumor microenvironment but exhibited pronounced heterogeneity, posing challenges for conventional clustering methods. By integrating genetic and metabolic features, we revealed key evolutionary trajectories of epithelial cancer cells during metastasis. Notably, we identified a novel epithelial subpopulation, NEndoCs, characterized by unique differentiation patterns and distinct spatial distribution across metastatic niches. We also observed significant metabolic reprogramming and recurrent mutations linked to prostate-to-bone microenvironmental transitions. **Conclusions:** This study comprehensively elucidates the mutation patterns, metabolic reprogramming, and microenvironment adaptation mechanisms of bone metastasis in prostate cancer, providing key molecular targets and clinical strategies for the precise treatment of bone metastatic prostate cancer.

## 1. Introduction

Prostate cancer (PCa) is one of the leading malignant tumors in terms of morbidity and mortality among men worldwide, particularly in aging populations, placing an increasing burden on healthcare systems [1,2,3]. Although localized prostate cancer is effectively treated with surgery and radiotherapy, approximately 20–30% of patients eventually develop distant metastases, with the bone as the predominant metastatic site [4,5,6]. Bone-metastatic prostate cancer substantially reduces patient survival, causes severe bone pain and fracture-related complications, thereby markedly impairing quality of life, which has become one of the important challenges in prostate cancer treatment [7,8].

The development of bone metastasis is a complex biological process involving interactions between cancer cells and the bone marrow microenvironment. Bone metastasis in prostate cancer is often accompanied by complex molecular alterations, including genetic mutations, epigenetic reprogramming, and clonal evolution, which govern the adaptive and evolutionary trajectories of tumor cells in the bone marrow microenvironment [9,10,11]. The heterogeneity of cancer cells renders traditional population-level analyses insufficient to fully capture their changes and diversity during bone metastasis. Therefore, employing high-resolution single-cell technologies to investigate distinct cell populations and molecular characteristics within the tumor microenvironment is crucial for elucidating the mechanisms of cancer metastasis.

With advances in high-throughput sequencing technology, single-cell transcriptomics has become a powerful tool for investigating the tumor microenvironment and tumor heterogeneity [12,13,14]. Analysis of single-cell transcriptome data elucidates tumor cell gene expression patterns across different developmental stages and uncovers the molecular basis of tumor metastasis, with a particular focus on prostate cancer bone metastasis. Single-cell technology enables the analysis of the tumor microenvironment and metastatic pathways at the single-cell level, offering novel insights into bone metastasis [15,16,17]. In addition, single-cell metabolic and gene mutation analyses further elucidate how cancer cells adapt to new microenvironments through metabolic reprogramming and genetic mutations during metastasis. Although single-cell transcriptomics has been applied in prostate cancer research, few studies have specifically investigated bone metastases, particularly lacking large-scale, high-quality integrated analyses of single-cell data.

This study aimed to systematically investigate cellular heterogeneity, functional dynamics, and microenvironmental adaptation mechanisms during prostate cancer bone metastasis through the integration of single-cell transcriptome data from primary and bone metastasis samples. We integrated data from 124 prostate and bone metastatic tissue samples, encompassing normal hip replacement tissue, benign prostatic hyperplasia, primary prostate cancer, bone metastatic prostate cancer, peripheral metastatic prostate cancer, distal bone metastatic prostate cancer, and bone marrow interstitial tissue. The final dataset comprised 602,497 high-quality single cells, representing the largest single-cell prostate cancer dataset reported to date, encompassing the most extensive sample collection and the greatest diversity of tissue types. Utilizing this dataset, we comprehensively analyzed cancer cell heterogeneity, metabolic reprogramming, gene mutations, and their interactions with the bone metastasis microenvironment. Furthermore, we delineated the clonal evolutionary trajectory of bone metastasis in prostate cancer and identify key genes and signaling pathways that may drive metastatic progression. Our findings offer novel insights into the occurrence and progression of bone metastases and establish a theoretical framework for the development of future precision treatment strategies.

## 2. Result

### 2.1. Sample Distribution and Data Integration in the Spatiotemporal Landscape of Prostate Cancer Bone Metastasis

To comprehensively characterize the single-cell transcriptomic features of prostate cancer bone metastases, we integrated and analyzed 124 scRNA-seq samples from public databases. These samples included prostate tissues representing diverse clinical conditions, including normal tissues from hip replacement patients, benign prostatic hyperplasia, primary prostate cancer, bone metastatic prostate cancer, peripheral metastatic prostate cancer, distal bone metastatic prostate cancer, and bone marrow stromal tissues. This study is the largest single-cell prostate cancer sample set reported to date, covering the largest number of samples and the richest sample types (Figure 1A). Among the 655,849 single cells initially obtained, we applied rigorous quality control based on the total number of transcripts, the number of detected genes, and the proportion of mitochondrial, hemoglobin, and ribosomal gene expression. As a result, we retained 602,497 high-quality cells for subsequent single-cell transcriptome analysis (Figure 1B). A total of 53,352 cells were filtered out, accounting for more than 8.13% of the initial dataset, demonstrating the most rigorous quality control process reported to date. Based on UMAP dimensionality reduction analysis, different cell populations formed distinct clusters in two-dimensional space, with a total of 60 cell clusters corresponding to specific cell subpopulations (Figure 1C). Further annotated analysis examined the influence of individual origin, sample type, and transition status. Cells from healthy, benign, and malignant tissues exhibited distinct distribution patterns, indicating differences in their transcriptional and functional characteristics (Figure 1D–F). The findings of this study provide new insights into disease progression and its microenvironment and lay the foundation for the development of personalized treatment strategies.

### 2.2. Single-Cell Landscape of Human Metastatic Prostate Cancer

Using the Seurat V5 tool [18], we classified the cells into nine categories based on their expression profiles of characteristic marker genes: epithelium cells, (NK)/T cells, fibroblasts, B cells, macrophages, endothelial cells, mast cells, and neutrophils (Figure 2A). The results of the cell composition analysis showed that epithelium cells and NK/T cells were the predominant cell types across all samples, accounting for 213,465 (35.4%) and 214,851 (35.66%) cells, respectively. Other identified cell types included 34,403 fibroblasts (5.71%), 29,401 B cells (4.88%), 50,067 macrophages (8.31%), 35,246 endothelial cells (5.85%), 9519 mast cells (1.58%), and 15,545 neutrophils (2.58%) (Figure 2B). To systematically explore the molecular characteristics of different cell populations, we used bubble plots and the FeaturePlots method to illustrate the expression patterns of specific marker genes. The epithelium cell marker genes included *EPCAM*, *KRT7*, *KRT8*, *KRT18*, and *KRT19*. The T/NK cell marker genes were *CD3D*, *CD3E*, *CD2*, *TRAC*, *TRBC1*, and *TRBC2*. The fibroblasts marker genes included *ACTA2*, *TAGLN*, *MYLK*, *MYL9*, *PDGFRB*, *NOTCH3*, *COL1A1*, and *DCN*. The macrophage marker genes were *CD68*, *CD168*, *CD14*, *C1QC*, *MRC1*, *CD1C*, *LAMP3*, *CSF3R*, *SORL1*, and *S100A8*. The neutrophil marker genes included *FCN1*, *CSF3R*, *SORL1*, and *S100A8*. The B cell marker genes were *MS4A1*, *CD79A*, *CD79B*, *MZB1*, *IGHG1*, and *JCHAIN*. The endothelial cell marker genes included *PTPRB*, *PECAM1*, and *RAMP2*. The mast cell marker genes were *KIT*, *ENPP3*, and *CPA3*. Furthermore, we conducted a further analysis of the composition of cell types in each dataset (Figure 2C and Appendix A). A more detailed comparative analysis revealed marked differences in cell composition between primary prostate cancer and bone metastatic prostate cancer, particularly concerning the relative abundance of epithelial and immune cells (Appendix A). Despite inter-individual and tissue-type biological heterogeneity, most samples exhibited a relatively stable cell composition pattern, which reflected the unique cellular ecosystem of prostate cancer (Figure 2D). In summary, this study established a comprehensive cellular atlas of prostate cancer bone metastasis based on a large-scale single-cell transcriptomic dataset.

### 2.3. Heterogeneity and Biological Characteristics of Epithelial Cells in Prostate Cancer

Cancer cells are the major component of the tumor microenvironment. However, due to the strong heterogeneity of cancer cells, cluster analysis has remained a challenge. To study the heterogeneity of PCa cells, epithelial cells were extracted for high-resolution analysis. Eight tumor cell subtypes were identified: proliferative (ProlifCs), metastatic (MetaCs), invasive (InvCs), immunogenic (ImmuCs), epithelial–mesenchymal transition (EMTCs), neuroendocrine (NEndoCs), osteoclastic (OstcCs) [19], and osteoblastic (OstbCs) [20,21] (Figure 3A). Marker gene analysis revealed distinct expression profiles for each subtype. For example, proliferative cells are characterized by the expression of *CD24*, *CCND1*, and *PCNA*, while MetaCs exhibit high levels of *TMSB4X*, *MDK*, and *CXCR4* (Figure 3B). These unique expression patterns characterize the functional transcriptomic landscape of epithelial subtypes. Copy number variation analysis using InferCNV revealed significant chromosomal alterations, including deletions on chromosome 6 (Figure 3C). Most epithelial cells showed significant CNV compared with normal cells, supporting their tumorigenic potential (Appendix A). In addition, the PCa scores of epithelial cells were significantly elevated compared with other cell types in the tumor microenvironment (TME), especially in prostate and bone tumor regions (Figure 3D). To further understand the functional role of epithelial subtypes, we used the Human Protein Atlas database [22] to conduct enrichment analysis of key marker genes (minimum expression fraction > 35%). The results revealed significant enrichment of marker genes for most epithelial subtypes in human prostate tissue (Appendix A). Eight tumor cell subtypes exhibited different transcriptional signatures and varied in proportion between the primary and bone metastatic samples (Figure 3E). These subtypes demonstrate different biological processes depending on their differential expression patterns (Figure 3F and Appendix A). These findings underscore the functional diversity of epithelial cell subtypes and their pivotal role in prostate cancer progression, providing valuable insights into tumor heterogeneity and potential therapeutic targets.

### 2.4. Inferring Metabolic Heterogeneity of Cancer Cells from scRNA-seq Data

Analysis of EMT scores among tumor cell subtypes [23] revealed that EMTCs exhibited the highest EMT scores, whereas OstcCs had the lowest, suggesting subtype-specific EMT characteristics. NEndoCs were highly enriched in bone metastases (Appendix A). The cellular composition of the tumor microenvironment (TME) plays a crucial role in the progression of prostate cancer and bone metastasis [24]. Given the strong association between prostate cancer and ChrY-linked genes, we analyzed the differential expression of ChrY-linked genes across various tissues and validated the findings using the TCGA male pan-cancer dataset (Appendix A). Our analysis revealed that *UTY* and *USP9Y* were highly expressed at the single-cell level and were significantly associated with bone metastasis (Appendix A). *UTY* and *USP9Y* are involved in demethylation and deubiquitination [25], respectively, and may potentially promote bone metastasis by modulating the immune microenvironment [26,27]. To further investigate the underlying mechanisms, we analyzed gene expression associated with immune and metabolic activity across different PCa regions [28]. Prostaglandin E2 (PGE2) inhibits T cell activation, thereby facilitating immune escape. In addition, PCa cells may promote T cell depletion by upregulating the expression of the taurine transporter *SLC6A6*, resulting in the competitive uptake of taurine (Appendix A). We analyzed the expression of immune- and metabolism-related genes and found that PTGES2 expression was reduced in bone metastases, whereas PTGES3 expression was significantly increased (Appendix A). TCGA data further indicate that high expression of *SLC6A6* and *PTGES3* is associated with *UTY* and *USP9Y*, potentially contributing to immune escape and promoting bone metastasis (Appendix A). Additionally, we identified key genes co-expressed with *UTY* and *USP9Y*, offering insights into potential therapeutic targets for bone metastasis in prostate cancer [29] (Appendix A).

Following this, the developmental trajectory and metabolic reprogramming characteristics of prostate cancer cells in different microenvironments were systematically analyzed through single-cell trajectory inference and metabolic analysis (Figure 4A). Trajectory inference results indicated that prostate cancer cells differentiated along two main trajectories: PC (prostate cancer) and BM (bone marrow microenvironment), with subsets of cells exhibiting high mutation burden gradually becoming the dominant subtypes. Including NEndoCs, OstbCs, ImmuCs, and InvCs, we found that *UTY*, *USP9Y*, and *BIRC6* exhibited a sustained pattern of high expression during bone metastasis, emphasizing their potential significance in bone metastasis development. Additionally, *CXCR4*, *IL17RA*, and *EMT* scores showed a phased pattern, initially increasing and then decreasing during bone metastasis. These trends, which first increased and then decreased during bone metastasis, suggest that these genes may play an important regulatory role in different stages of bone metastasis (Figure 4B and Appendix A). Simultaneously, using lattice heat maps of gene expression correlations, we demonstrate the cooperative or mutually exclusive expression relationships of different genes in this evolutionary trajectory (Appendix A). To deepen the understanding of metabolic differences between bone metastatic cancer and prostate cancer, we employed a detailed neural network model to infer single-cell metabolism [30]. Metabolic pathway analysis revealed that different epithelial cell subtypes exhibited significant metabolic characteristics overall. For example, OstbCs and OstcCs, which are closely related to bone metastasis, exhibit increased activity in fatty acid metabolism and steroid hormone synthesis (Figure 4C). Further metabolic profiling revealed significant changes in glutamine, cholesterol, and related amino acid metabolites in prostate cancer cells and bone marrow microenvironments, and these metabolites may play a key role in tumor progression and microenvironment adaptation (Figure 4D). Beyond metabolic activity, prior studies have demonstrated that distinct expression profiles of cytokines and their receptors play a crucial role in TME remodeling [31,32]. Our data revealed higher cytokine expression in PCa samples than in bone tissue (Appendix A).

### 2.5. Single-Cell Mutation Analysis Reveals the Microevolutionary Process of Cancer Cells

The microevolution of cancer cells refers to the accumulation of genetic and epigenetic variations during tumorigenesis and progression, leading to the formation of sub-clonal populations with distinct biological features [33,34]. Investigating the microevolution of cancer cells is crucial for understanding tumor biology. To explore the clonal evolution pattern of PCa cells during bone metastasis, we conducted somatic mutation analysis at the single-cell level and characterized the mutation landscape across tumor subtypes through a comprehensive analysis of tumor gene mutation data. First, among the 505 samples, 225 (50.5%) harbored gene mutations, such as missense mutations, nonsense mutations, and splicing site mutations, with certain genes exhibiting high mutation frequencies (Figure 5A). Next, base substitution analysis revealed that T > C transitions were the most prevalent base substitutions, with a distinction between transitions (Ti) and transversions (Tv) (Figure 5B). We further quantified the mutational burden across epithelial subtypes and identified a stepwise increase from early-stage to more aggressive subtypes, such as InvCs and ImmuCs (Figure 5C). To study the hierarchical organization of these subpopulations, we constructed a pseudotemporal developmental tree to reveal evolutionary trajectories and potential transitions among subtypes. Notably, two distinct populations of ImmuCs were observed at the first branching point, indicating functional heterogeneity within the immune microenvironment. Although both cell populations are classified as immune cells, they may represent different functional states, such as pro-inflammatory and immunomodulatory states, suggesting distinct immune responses throughout disease progression (Figure 5D). Additionally, mutation burden analysis revealed significant variations among tumor subtypes, with certain subtypes exhibiting high mutation burdens, potentially associated with genomic instability [35]. UTY exhibited the highest mutation burden among all PCa cell subgroups (Appendix A). In summary, studying the microevolution of cancer cells is essential for revealing tumor heterogeneity, understanding tumor progression mechanisms, advancing precision medicine, fostering therapeutic innovations, and deepening our understanding of the tumor microenvironment.

### 2.6. Cell Mutation Analysis Reveals the Mutation Landscape of Cancer Cells

To better understand how these mutation patterns influence the initiation, progression, and bone metastasis of prostate cancer cells, we defined six functional biological modules: transcriptional regulation, signal transduction, extracellular matrix organization, circadian rhythm, metabolism, and prostate cancer-specific functions. Additionally, we defined a separate module composed of ChrY genes. Based on the signature genes and tumor cell subtypes in the samples, we constructed pseudo-batch data and calculated the mutation rates of these functional biological modules (Figure 6A). Using the male pan-cancer dataset from TCGA, we found that the mutant group exhibited worse prognosis than the wild-type group, thus validating the clinical and biological relevance of these functional modules in cancer research (Appendix A). At the single-cell transcriptome level, we found that mutations in functional biological modules significantly altered transcriptomic expression patterns. Except for the circadian module, the expression levels of the remaining five modules were elevated in the mutant group compared to the wild-type group (Figure 6B and Appendix A). In addition, InvCs and MetaCs cells were predominantly classified within the wild-type group, indicating that certain PCa cell subtypes possess the capability for bone metastasis and invasion. This process is potentially mediated through the regulation of mutation patterns and circadian rhythm-associated gene expression levels (Appendix A).

## 3. Discussion

In this study, single-cell RNA sequencing (scRNA-seq) was employed to generate a comprehensive cellular atlas of prostate cancer (PCa) bone metastases, revealing the heterogeneity of cell types and their associated molecular mechanisms. Specifically, we identified eight epithelial cell subtypes, including ProlifCs, MetaCs, InvCs, ImmuCs, EMTCs, NEndoCs, OstcCs, and OstbCs. These subtypes exhibit distinct transcriptomic features that highlight the functional heterogeneity of prostate cancer bone metastases. Our findings indicate that PCa cells adapt to the bone microenvironment by reprogramming their metabolism and upregulating growth-promoting pathways. This is consistent with previous studies suggesting that tumor cells exploit lipid metabolic pathways to regulate immunosuppression as part of their survival strategy within the bone microenvironment [36,37,38].

In addition to metabolic and immune adaptation, we identified NEndoCs as a distinct and functionally significant epithelial cell subtype. NEndoCs were preferentially enriched in bone metastatic lesions and displayed a distinct transcriptomic profile, suggesting their development into a specialized cell state during late-stage disease progression. Pseudo time trajectory analysis showed multiple evolutionary branches converging onto the NEndoCs population with high transition probabilities, supporting a model of convergent differentiation. This suggests that NEndoCs may originate from multiple tumor cell lineages in response to selective pressures. Given the established role of neuroendocrine differentiation in therapeutic resistance and aggressive tumor progression, the presence of NEndoCs suggests a potential role for lineage plasticity in prostate cancer bone metastasis. These findings necessitate further investigation to elucidate the molecular mechanisms driving NEndoCs formation and their clinical significance.

Functional pathway analysis revealed significant upregulation of lipid metabolism and energy production pathways in bone metastatic samples. Concurrently, immune evasion and microenvironmental adaptation pathways, including TGF-β signaling and immunosuppressive mechanisms, were significantly upregulated. These observations support the hypothesis that tumor cells undergo metabolic adaptation through bidirectional interactions with the bone microenvironment, facilitating their successful colonization and dissemination [39,40,41]. In addition, our study identified two Y chromosome-linked genes, *UTY* and *USP9Y*, which are highly expressed in PCa bone metastases and are strongly associated with tumor cell proliferation, metastasis, and immune evasion. Through pseudo-time analysis, we traced the dynamic trajectory of tumor cells from primary prostate cancer to bone metastasis, highlighting the crucial role of *UTY* and *USP9Y* in this process. Our data indicate that these genes may promote prostate cancer bone metastasis by modulating the tumor immune microenvironment and metabolic pathways. Both genes are located on chromosome Yq11.2 and may contribute to prostate cancer progression by modulating androgen signaling pathways [42,43]. UTY encodes a histone demethylase that is capable of epigenetically regulating AR (androgen receptor) target genes such as PTEN, thereby potentially promoting tumor cell proliferation [44]. *USP9Y* acts as a deubiquitinase and may stabilize AR and its coactivators, thereby enhancing AR signaling [45]. Dysregulation or deletion of these genes, often observed in Y chromosome loss, has been linked to the aberrant activation of AR splice variants (e.g., AR-V7) and the NF-κB pathway, both of which contribute to the transition to castration-resistant prostate cancer [46,47]. This finding offers novel biomarkers and potential therapeutic targets for elucidating the mechanisms underlying prostate cancer bone metastasis. In summary, this study uncovers the cellular heterogeneity and the molecular underpinnings of bone metastasis in prostate cancer through single-cell transcriptomics.

However, this study has several limitations. First, although we employed large-scale single-cell RNA sequencing data, our analysis was primarily centered on gene expression profiles of tumor cells and their immediate microenvironment, providing only limited insights into other regulatory factors, such as the influence of the remote microenvironment. Second, while our findings indicate that *UTY* and *USP9Y* are critically involved in bone metastasis, the precise mechanisms underlying their function remain unclear. Future research should aim to elucidate the mechanisms by which these genes contribute to tumor metastasis through the regulation of immune evasion, metabolic pathways, and cell–cell interactions. First, the deubiquitinating activity of *USP9Y* on AR or its coactivators (e.g., SRC-1) can be examined using co-immunoprecipitation (Co-IP) combined with ubiquitination assays to evaluate its role in protein stabilization. Second, in vitro functional studies, particularly under androgen deprivation therapy (ADT) conditions, can clarify how *UTY* or *USP9Y* loss affects tumor cell proliferation, apoptosis, invasion, and organoid-forming capacity. Third, potential pharmacological combination strategies could be investigated; for example, combining the *USP9Y* inhibitor EOAI3402143 with enzalutamide could potentiate cytotoxic effects in castration-resistant prostate cancer (CRPC) models. Lastly, high-throughput epigenetic drug screening could be utilized to assess the selective efficacy of histone demethylase inhibitors, such as GSK-J4, in tumors exhibiting high UTY expression. These experimental approaches will play a crucial role in elucidating the functional relevance of *UTY* and *USP9Y* and in developing potential therapeutic strategies for targeting these Y chromosome-linked genes.

## 4. Methods

### 4.1. Patient Sample Collection

Single-cell RNA sequencing data were obtained from publicly available prostate cancer samples in the Gene Expression Omnibus (GEO) database (https://www.ncbi.nlm.nih.gov/geo, accessed on 1 December 2023). All authors have agreed to the submission of this manuscript, ensuring the accuracy and completeness of the data presented. The scRNA-seq datasets used in this study are accessible under the following GEO accession numbers: GSE137829, GSE143791, GSE181294, GSE193337, GSE206962, GSE244267, GSE157703, and GSE166782.

### 4.2. Single-Cell RNA Sequencing

For each sample, the raw data were analyzed using the Cell Ranger (Version 7.1.0) pipeline, with gene count data generated using the default and recommended parameters. FASTQ output obtained from sequencing data was aligned with the GRCh38 reference genome using the STAR algorithm. We used Seurat (Version 5.0.1) R package to do all subsequent analyses [18]. Cells with <200 features or >10,000 features or >20% reads mapped to mitochondrial genes were filtered out from the downstream analysis. We used Seurat::NormalizeData, Seurat::FindVariableFeatures and Seurat::ScaleData functions to normalize all data. The top 20 PCS were used in Seurat::RunUMAP visualization. Cluster analysis was performed using the Seurat::FindNeighbors and Seurat::FindClusters functions. In this study, a two-step clustering method was used to identify the major and minor subtypes of cells. First, the data were initially processed using the single-cell data processing process described above and unsupervised clusters were obtained. The Seurat::DotPlot function was then used to visualize the major marker genes for each cluster and annotated manually in turn.

### 4.3. Cell Type Annotation

In this study, a two-step clustering method was used to identify the major and minor subtypes of cells. First, the data were initially processed using the single-cell data processing process described above and unsupervised clusters were obtained. The Seurat::DotPlot function was then used to visualize the major marker genes for each cluster and annotated manually in turn. The marker genes used in the major cell types were as follows: epithelium cell (*EPCAM*, *KRT7*, *KRT8*, *KRT18*, *KRT19*), T/NK cell (*CD3D*, *CD3E*, *CD2*, *TRAC*, *TRBC1*, *TRBC2*); fibroblast (*ACTA2*, *TAGLN*, *MYLK*, *MYL9*, *PDGFRB*, *NOTCH3*, *COL1A1*, *DCN*); macrophage (*CD68*, *CD168*, *CD14*, *C1QC*, *MRC1*, *CD1C*, *LAMP3*, *CSF3R*, *SORL1*, *S100A8*); neutrophil (*FCN1*, *CSF3R*, *SORL1*, *S100A8*); B cell (*MS4A1*, *CD79A*, *CD79B*, *MZB1*, *IGHG1*, *JCHAIN*); endothelium (*PTPRB*, *PECAM1*, *RAMP2*); and mast (*KIT*, *ENPP3*, *CPA3*). The minor subtypes are annotated sequentially from the well-annotated major subtypes, using the same single-cell sequencing analysis workflow [31,48,49,50,51,52,53].

### 4.4. Copy Number Variation Analysis

The CNV result-based scRNA-seq data were performed by inferCNV (Version 1.20.0) R package (https://github.com/broadinstitute/infercnv, accessed on 12 March 2024). Epithelium cells were used for the observations and all other cells as references. Cut-off for the min average read counts per gene among reference cells (cutoff = 0.1). Due to the huge data resources used in this study, inferCNV was unable to calculate all the cells at once. Therefore, for the sake of visualization, we randomly down-sampled the epithelium cells and normal cells in each group. There were 12,000 epithelial cells and 8000 normal cells in each group. Then, we used the output of inferCNV to further evaluate the copy number variation scores in the epithelial cells. First, we read and loaded the processed inferCNV object. Then, we extracted the expression data from it and annotated the gene locations. Next, we grouped genes ChrY and calculated the *CNV scores* for each chromosome. Based on the calculation results, we further evaluated the *CNV scores* for each cell and jointly analyzed them with the metadata to identify potential malignant cells. The formula for calculating cell *CNV score* is as follows:$${CNV\_score}_{j}=\frac{1}{N}\sum_{i}^{n}{\left({expr}_{ij}-1\right)}^{2}$$

### 4.5. Calculation of Gene Signature Scores

Using single-cell RNA sequencing data, we calculated the scores for multiple gene features. For each gene feature, we used the Seurat::FindAllMarkers function to score individual cells. The function calculates the average expression level of the selected gene at the single-cell level and subtracts the aggregated expression of the control feature set. For malignant cells, we calculated the prostate cancer score based on the expression of 19 genes related to prostate cancer (*KLK2*, *KLK3*, *AR*, *NKX3-1*, *TMPRSS2*, *HOXB1*, *EPCAM*, *KRT8*, *KRT18*, *KRT5*, *KRT15*, *KRT17*, *FOLH1*, *MSMB*, *NEFH*, *AZGP1*, *RDH11*, *PLA2G2A*, and *FOLH1*). Meanwhile, we calculated the EMT score of prostate cancer cells based on the expression of eight genes related to epithelial–mesenchymal transition (*CDH2*, *VIM*, *SNAI1*, *SNAI2*, *TWIST1*, *MMP2*, *MMP3*, *MMP9*) [23]. For the different subtypes of TME, differential genes between subtypes were identified using the Seurat::FindMarkers function, and genes of interest were selected with a logFC value greater than 1 and a minimum expression proportion greater than 0.35, including 30 genes related to BM comrades (*APOE*, *ASPN*, *BGN*, *C4orf48*, *COL12A1*, *COL1A1*, *COL1A2*, *COL3A1*, *COL4A1*, *COL5A1*, *COL5A2*, *COL8A1*, *CRIP2*, *CTHRC1*, *FPR3*, *GPNMB*, *INHBA*, *MRC1*, *MSR1*, *NRP1*, *OLFML2B*, *OLR1*, *PDGFRB*, *POSTN*, *SFRP4*, *SLCO2B1*, *SPARC*, *STPP1*, *THBS2*, *THY1*), and 17 genes related to PC comrades (*APOE*, *BGN*, *COL1A1*, *COL1A2*, *COL3A1*, *COL4A1*, *CRIP2*, *FRZB*, *GJA4*, *IER5L*, *NDUFA4L2*, *NOTCH3*, *NRP1*, *PDGFRB*, *SPARC*, *TAF10*, *THY1*).

### 4.6. Pathway Enrichment Analyses

Gene ontology enrichment analysis (GO) [54]. To identify enriched Gene Ontology (GO) biological processes based on differentially expressed genes (DEGs), analysis was performed on DEGs from each cell type using clusterProfiler::gseGO (Version 4.8.1) R package, using org.Hs.eg.db annotation Package (Version 3.17.0), and functional enrichment analysis and cell markers annotation enrichment. The gene lists were submitted to the EnrichR (https://maayanlab.cloud/Enrichr/, accessed on 22 April 2024) online tool and the top ten terms were retained according to the adjusted *p*-value.

### 4.7. Single-Cell Level Metabolic Analysis

We used the scFEA [30] algorithm to estimate the metabolic level of each cell based on scRNA-seq data. First, the original expression counts were normalized to counts per million (CPM), which were then used as input for the python package scFEA. The default parameters were used to infer the metabolic flux values of 168 core metabolic reactions implemented in scFEA. The inferred metabolic flux values were compared between paired mutant and wild-type samples using Seurat::FindMarkers function with the option “LR” (test. use = ’LR’) for logistic regression and likelihood ratio tests, with experimental batch information included as a covariate. Constant low-flux reactions with differences < 1.0 × 10^−4^ between minimum and maximum fluxes were excluded. In the visualization stage, we grouped cells based on the grouping information provided in the metadata and calculated the average metabolic expression level of all cells within each group. We then conducted differential analysis of the metabolic expression matrix using the Wilcoxon rank-sum test. Differences were considered statistically significant at *p* < 0.05, and the tendency of differential expression was assessed using Cohen’s d index.

### 4.8. De Novo Detection of Somatic Mutations

The cell-type level somatic mutation analysis was performed using the SComatic [35] protocol. First, we labeled the cell type for each cancer cell and outputted the results in a list format (the first column representing the cell barcode and the second column representing the cell identity). Then, we split the BAM files from scRNA-seq into separate files based on cell identity. Next, we ran SComatic to generate a VCF file for downstream analysis. When collecting base count information, we filtered out data with a mapping quality below 255 or more than 5 mismatches per read. Additionally, we used whole-genome sequencing (WGS) data from the 1000 Genomes Project to generate a Panel of Normals (PON) as a reference dataset. The final inferred results were intersected with high-quality regions of the human genome. All output VCF files were annotated using snpEff software (version 5.2c), which utilizes the GRCh38 reference genome file.

### 4.9. Pseudo Trajectory Inference Analyses

We used Slingshot [55] (version 2.7.0) for time trajectory inference. For T cells, we took native T cells as the starting point for the time trajectory; for cancer cells, we took rolifCs as the starting point. For each analysis, the single-cell analysis workflow was performed on both cell types, followed by dimensionality reduction based on PCA, and then visualized in 2D on UMAP. During trajectory inference, multiple differentiation trajectories may be obtained. In highly overlapping developmental trajectories, we extracted the main cell development path that was not overlapping and contained the largest number of cells as the inference result.

### 4.10. Pseudo-Evolution Tree

We performed pseudo-time analysis on malignant epithelial cells using the R package dyno [56] (Version 0.1.2). All parameters and settings were kept at their default values. Finally, we utilized PAGA-TREE [57] for pseudo-time analysis and constructing the pseudo-evolution tree. We used matplotlib in python for visualization.

### 4.11. Pseudo-Bulk Differential Expression Analysis

Utilizing Seurat::AggregateExpression for pseudo-bulk differential expression analysis, cell types were categorized, and the expression matrix for each group was aggregated and summarized to create a pseudo-bulk expression matrix based on the specified grouping criteria. Prior to identifying differentially expressed genes, the pseudo-bulk expression matrix underwent preprocessing in accordance with the previously outlined single-cell data processing workflow. Differential gene expression was then determined using Seurat::FindMarker. Genes exhibiting a LogFC of at least 1, a *p*-value after Benjamini-Hochberg (BH) correction of less than 0.05, and a minimum expression proportion exceeding 0.35 were deemed to have statistically significant differential expression.

### 4.12. TCGA Data Analysis

We utilized the R software package easyTCGA (Version 0.0.4.200.0) to download TCGA pan-cancer gene expression matrices and sample clinical information for analysis of transcriptome expression matrix data. Additionally, we obtained mc3.v0.2.8.PUBLIC.maf.gz from the Scalable Open Science Approach for Mutation Calling of Tumor Exomes Using Multiple Genomic Pipelines for analysis of genomic mutation data. For summarizing, analyzing, annotating, and visualizing the mutation data, we used maftools (Verison 2.20.0). Finally, Kaplan–Meier survival curves were plotted using the ggsurvplot function in the R package Survminer (Version 0.4.9).

## 5. Conclusions

By integrating single-cell transcriptome data, this study systematically elucidated the cellular heterogeneity, metabolic reprogramming, and microenvironment adaptation mechanisms of prostate cancer during bone metastasis. The findings indicate that NEndoCs cancer cell subpopulations exhibit unique differentiation pathways within the bone metastasis microenvironment, highlighting their critical role in prostate cancer bone metastasis. In addition, large-scale single-cell atlas analysis elucidated the mutational pattern of bone metastasis in prostate cancer and its adaptive strategies within the tumor microenvironment, providing potential molecular targets for future precision therapy. The results of this study offer new insights into the mechanisms underlying bone metastasis in prostate cancer and provide theoretical support for the development of clinical targeted therapy strategies.

## Figures and Tables

**Figure 1 biomedicines-13-00909-f001:**
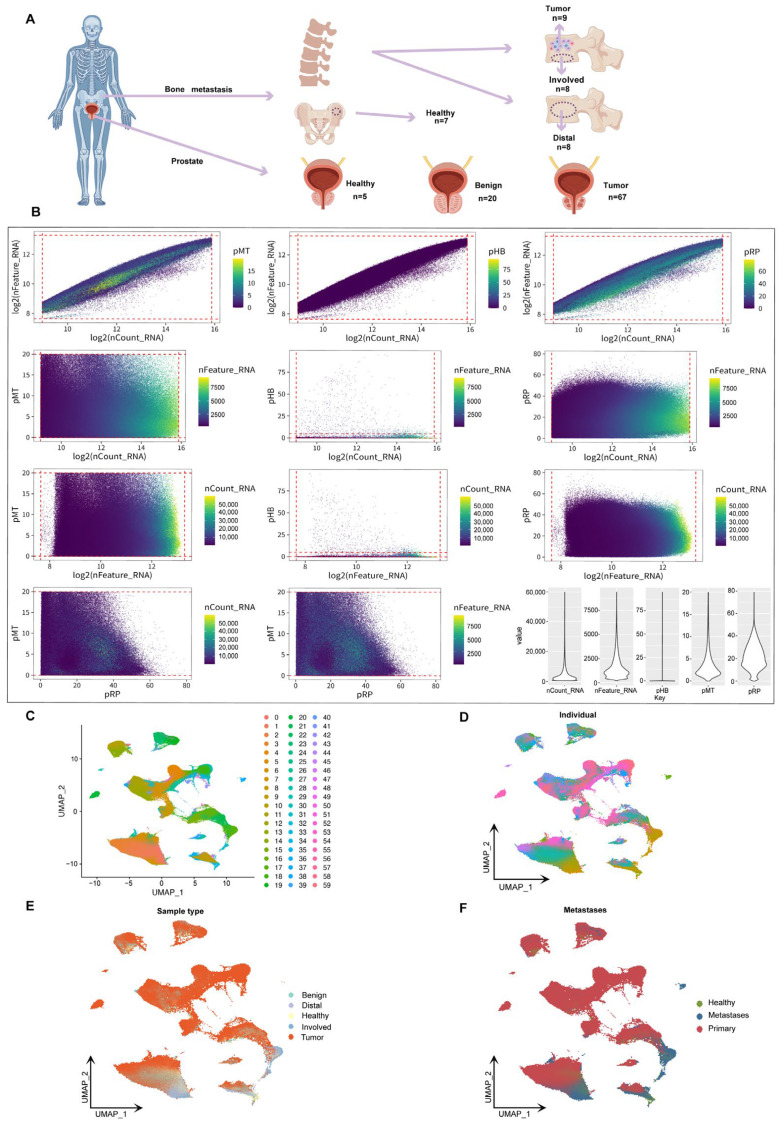
**UMAP analysis and classification of prostate cancer bone metastases.** (**A**) Schematic showing sample sources, including prostate tissues (healthy, benign, tumor) and bone marrow samples from tumor sites (Involved), distant vertebrae (Distal), and hip-derived healthy controls. (**B**) Pairwise plots of key quality control metrics: RNA counts (nCount_RNA), gene features (nFeature_RNA), and percentages of mitochondrial (pMT), hemoglobin (pHB), and ribosomal (pRP) genes. Violin plots (right) show distributions across all cells. (**C**) Shows the clustering of samples in two-dimensional space, with colors representing different clustering groups. (**D**) The UMAP shows the cell distribution of a single sample (*n* = 124). Each color represents a different sample to account for inter-individual differences in cell transcriptome profiles. (**E**) UMAP coloring analysis based on sample type (benign, healthy, tumor, Distal, Involved). (**F**) Displays the distribution of transition states (healthy, metastatic, primary) in the UMAP space.

**Figure 2 biomedicines-13-00909-f002:**
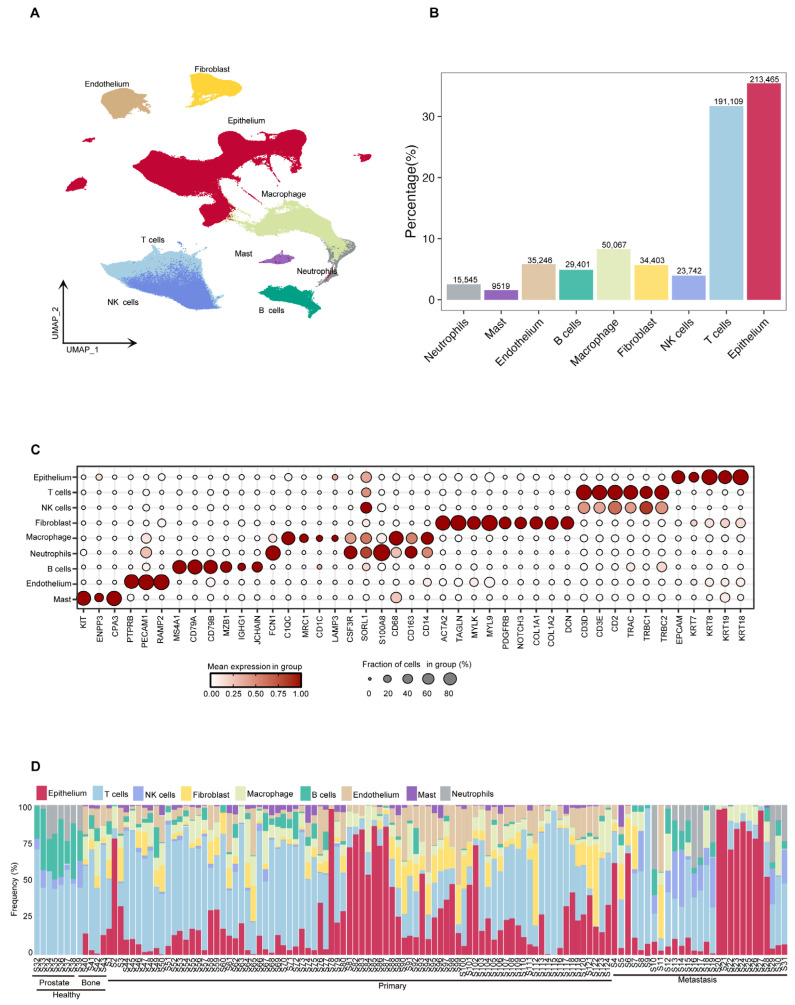
Analysis of prostate cancer cell composition, gene expression characteristics and inter tissue differences. (**A**) UMAP projection of all high-quality cells (n = 602,497), colored by major cell types, including epithelial cells, T cells, NK cells, B cells, myeloid cells (macrophages, monocytes), fibroblasts, and endothelial cells. (**B**) Bar plot showing the proportion and total number of each cell type across the dataset. (**C**) Dot plot showing the expression of selected marker genes (e.g., *EPCAM*, *KRT18*, *CD3D*, *CD79A*, *CD14*, *COL1A1*, *PECAM1*) across the identified cell types. Dot size indicates the fraction of cells expressing the gene, and color intensity reflects the average expression level. (**D**) Stacked bar plot showing the relative cell type proportions per sample, grouped by individual and tissue type. This reveals inter-sample and inter-tissue variation in cell composition.

**Figure 3 biomedicines-13-00909-f003:**
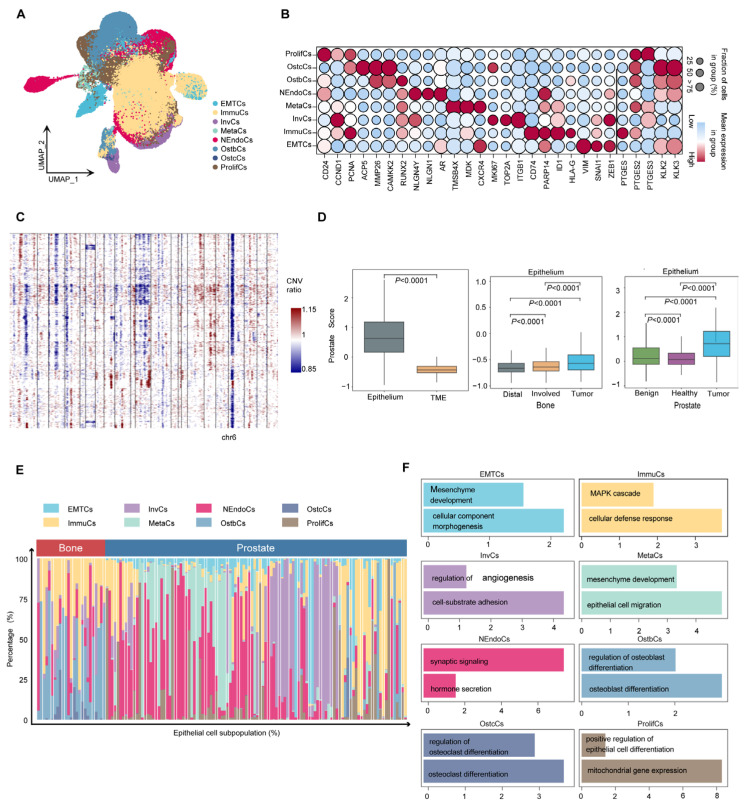
**Heterogeneity of prostate cancer cells.** (**A**) UMAP embedding of epithelial cells reveals eight transcriptionally distinct subpopulations, including ProlifCs, MetaCs, InvCs, ImmuCs, EMTCs, NEndoCs, OstcCs, and OstbCs states. (**B**) Dot plot showing the expression of representative marker genes across epithelial subtypes. Dot size indicates the percentage of cells expressing each gene; color intensity reflects average expression. (**C**) Copy number variation (CNV) heatmap across epithelial subtypes, inferred from averaged expression along the genome. Blue and red indicate relative loss or gain of genomic regions, respectively. (**D**) Box plots of epithelial versus tumor microenvironment (TME) scores across different tissues. Comparisons include tumor vs. TME, Distal vs. Involved bone marrow, and prostate tissue states (Healthy, Benign, Tumor). (**E**) Stacked bar plot showing the proportional distribution of epithelial subpopulations in prostate versus bone-derived tissues across all individuals. (**F**) Functional enrichment analysis of each epithelial subpopulation, highlighting major biological pathways such as epithelial differentiation, osteoclast/osteoblast regulation, immune response, and mesenchymal development.

**Figure 4 biomedicines-13-00909-f004:**
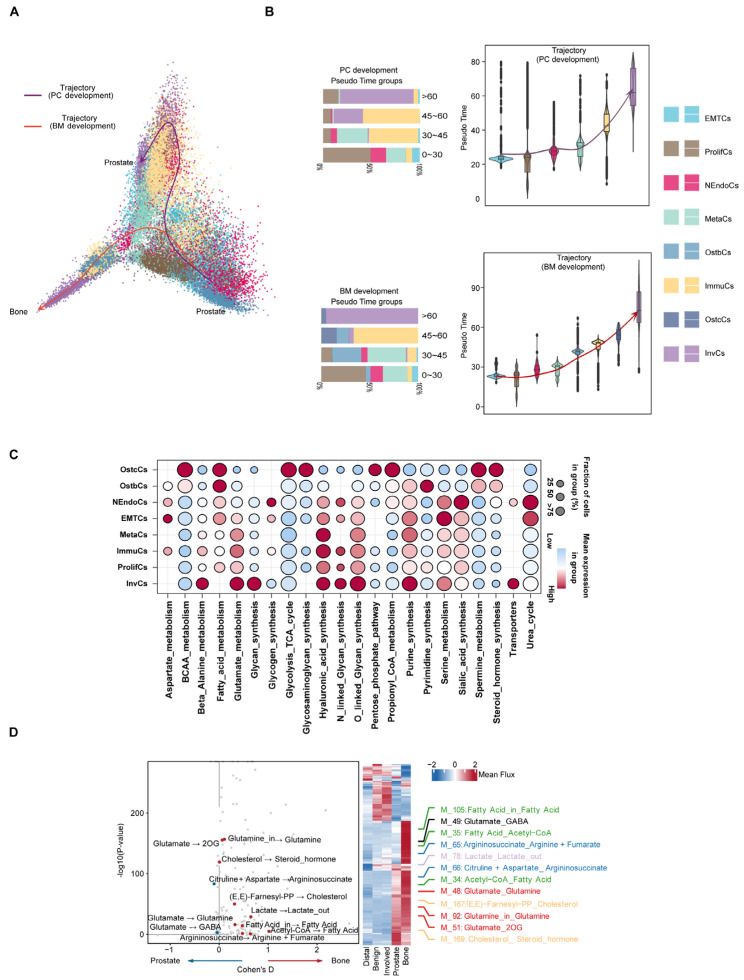
**Evolutionary trajectories and regulatory features of prostate cancer progression and bone metastasis.** (**A**) Pseudo-temporal analysis reveals two major developmental paths: prostate cancer progression (PC trajectory, purple) and bone metastasis adaptation (BM trajectory, red). Cells are colored by epithelial subtypes. Arrows indicate the inferred direction of differentiation across tumor states. (**B**) Violin plots showing the distribution of epithelial cell subtypes along pseudo time in both PC and BM trajectories. Bar plots on the left summarize the subtype composition in each pseudo time group. (**C**) Bubble plot showing enrichment of transcription factors (TFs) across epithelial subtypes, highlighting key regulatory programs associated with specific cell states. Dot size represents proportion of cells expressing the TF; color reflects average expression level. (**D**) Differential metabolic flux analysis between prostate and bone-derived epithelial cells. Left: volcano plot showing significantly altered metabolic reactions. Right: heatmap and line plots illustrate mean flux differences in key pathways (e.g., glutamine metabolism, lipid metabolism, steroid biosynthesis) across sample types.

**Figure 5 biomedicines-13-00909-f005:**
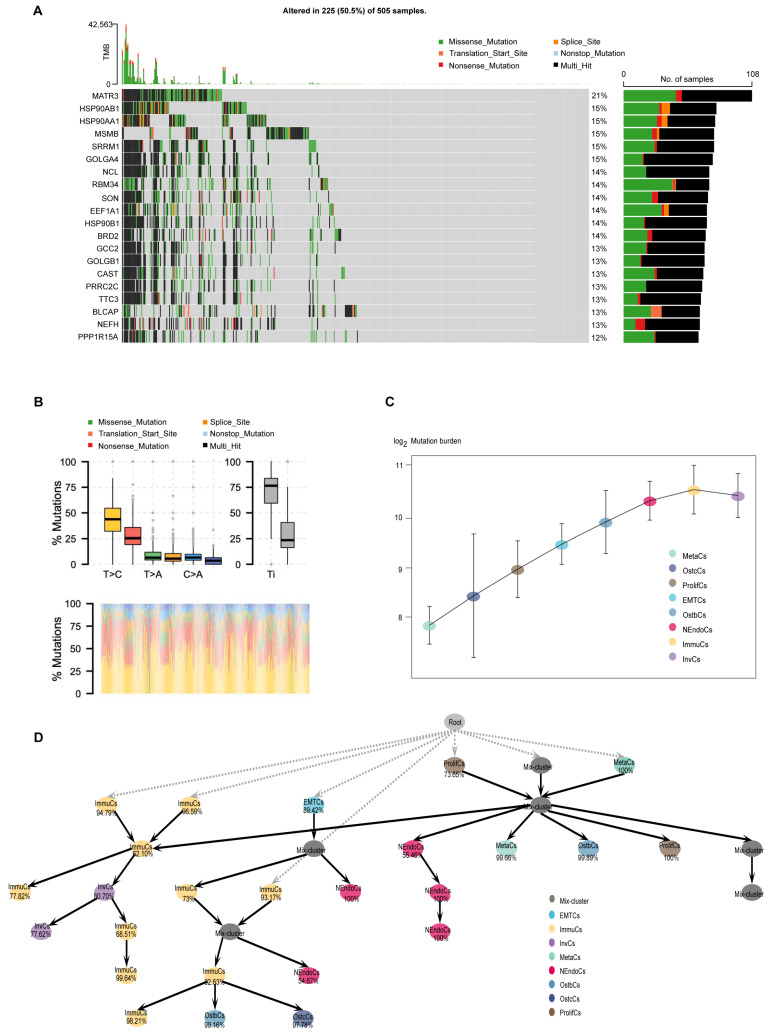
**Mutational landscape, regulatory networks, and gene expression heat maps of prostate cancer and its subtypes.** (**A**) Oncoplot showing somatic mutations across 505 prostate cancer samples. Frequently mutated genes are listed on the left, with mutation types color-coded (e.g., missense, nonsense, splice site). The bar plot on the right shows the mutation frequency of each gene. (**B**) Distribution of mutation types (e.g., T > C, C > A, T > A) across clinical groups. Lower panel shows mutational signature contributions by sample. (**C**) Mutation burden (log-transformed) across epithelial cell subtypes, indicating increased mutational accumulation along tumor progression. (**D**) Inferred developmental lineage tree of prostate cancer epithelial subpopulations. Nodes represent distinct subtypes or intermediate states, and arrows indicate potential transition paths. Percentages denote the confidence or assignment probability of each node. The tree structure suggests hierarchical progression from root-like proliferative or mixed states toward more specialized subtypes such as MetaCs, NEndoCs, or OstbCs.

**Figure 6 biomedicines-13-00909-f006:**
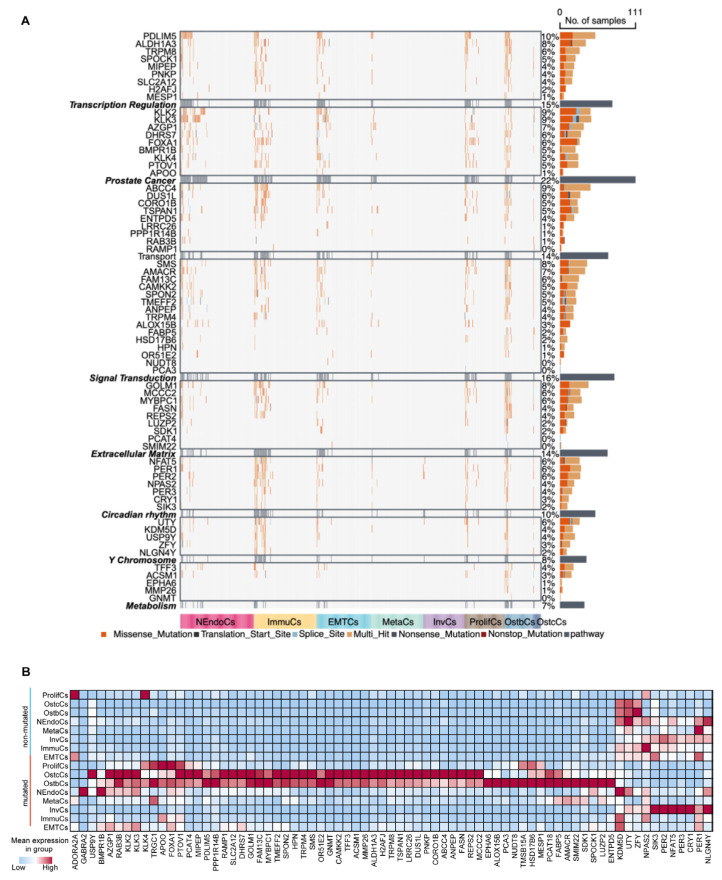
**Mutational landscape and pathway changes in prostate cancer development.** (**A**) Heatmap illustrating mutations in key genes across prostate cancer samples, categorized into functional groups such as transcriptional regulation, prostate-related pathways, signal transduction, extracellular matrix remodeling, circadian rhythm, Y chromosome-associated genes, and metabolism. The mutation types analyzed (e.g., missense, nonsense, splice site, and indels) are color-coded for visualization, and the number of samples harboring mutations in each gene is visualized as bar plots on the right. Subtypes are represented by distinct colors at the bottom of the heatmap. (**B**) Heatmap of gene expression across different prostate cancer subtypes. Red denotes high expression levels, whereas blue indicates low expression levels. Subtypes are ordered to emphasize differences in gene expression patterns, reflecting tumor heterogeneity.

## Data Availability

The data are from the GEO database public data set and can be downloaded directly.

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
