# Peer review of "Comprehensive Integrated Analysis Reveals the Spatiotemporal Microevolution of Cancer Cells in Patients with Bone-Metastatic Prostate Cancer"

_biomedicines, 2025, doi:10.3390/biomedicines13040909_

Round 1

Reviewer 1 Report

Comments and Suggestions for Authors

The manuscript “Comprehensive integrated analysis reveals the spatiotemporal microevolution of cancer cells in patients with bone-metastatic prostate cancer” from Feng Y. et al. submitted to Biomedicines is well written, and I want to congratulate the authors for this nice and clean submission.

Minor comment:

  1. Please check this sentence for clarity. “We evaluated EMT scores across tumor cell subtypes [23] and found that EMT scores 199 were the highest, whereas OSTCC scores were the lowest”. (Line 199).
  2. Figure 1B legend or main text should include a brief description as it is currently not clear from 1B how it depicts the quality control process. The preceding sentence states that 602,497 cells were included in the analysis out of an initial 655,849 cells.

Technical comments:

  1. From healthy control (patients undergoing hip replacement), what was the anatomic site of samples collected and analyzed? These samples (S32-S39) showed only immune cells, no epithelial cells at all. Figure 2D also shows 12 healthy control samples (S32-43). Please show sub-labels for normal prostate tissue and bone tissues under the healthy control label.
  2. While the study identifies UTY and USP9Y as critically involved in bone metastasis, the precise mechanisms underlying their function remain unclear. Could authors provide more clarity on these 2 Y chromosome localized genes? How are these involved in prostate cancer progression in the context of Androgen signaling, which plays an important role in prostate cancer progression? Please provide this discussion in the discussion section of the manuscript.
  3. The study relies heavily on computational analyses and public datasets and lacks experimental validation of key findings, such as the role of UTY and USP9Y in bone metastasis. I understand that validating all the findings would be an unfair expectation here. However, I still suggest that the authors include a paragraph in the discussion to highlight actionable directions clearly.
  4. Since this manuscript is heavy in bioinformatics analysis, I suggest that the authors add a glossary of terms for quick reference for the readers. A table of quick-referring reading text should explain the analysis in a few sentences, which will help readers with little or no bioinformatics background understand the analyses.
  5. Also, I would strongly recommend expanding legends to provide more information rather than just one-liner for each subfigure. Currently, it is difficult to understand many of the subfigures based on the information provided in the legends.

Author Response

Response to Reviewer X Comments

1. Summary

Thank you very much for taking the time to review our manuscript. We appreciate the reviewers’ thoughtful feedback and constructive comments, which have helped us improve the clarity and quality of our work. Please find below our detailed responses to each comment, and the corresponding revisions have been highlighted in yellow in the re-submitted manuscript. We have carefully addressed all concerns raised. Where we respectfully disagree, we have provided clear justifications and further clarification in our responses, and we welcome any additional guidance from the Academic Editor.

2. Questions for General Evaluation

Reviewer’s Evaluation

Response and Revisions

Does the introduction provide sufficient background and include all relevant references?

Yes/Can be improved/Must be improved/Not applicable

Yes

Are all the cited references relevant to the research?

Yes/Can be improved/Must be improved/Not applicable

Yes

Is the research design appropriate?

Yes/Can be improved/Must be improved/Not applicable

Yes

Are the methods adequately described?

Yes/Can be improved/Must be improved/Not applicable

Yes

Are the results clearly presented?

Yes/Can be improved/Must be improved/Not applicable

Yes

Are the conclusions supported by the results?

Yes/Can be improved/Must be improved/Not applicable

Yes

3. Point-by-point response to Comments and Suggestions for Authors

Minor comment:

Comments 1: Please check this sentence for clarity. “We evaluated EMT scores across tumor cell subtypes [23] and found that EMT scores 199 were the highest, whereas OSTCC scores were the lowest”. (Line 199).

Response 1: Thank you for pointing this out. We agree with your suggestion and have revised the sentence to improve clarity and accuracy. The updated sentence can be found in the revised manuscript on lines 211–213.

Comments 2: Figure 1B legend or main text should include a brief description as it is currently not clear from 1B how it depicts the quality control process. The preceding sentence states that 602,497 cells were included in the analysis out of an initial 655,849 cells.

Response 2: Thank you for this helpful comment. We agree with your suggestion. To improve clarity, we have revised the main text to include a concise description of the quality control process when referencing Figure 1B. In addition, a brief explanation has been added to the figure legend. These revisions can be found in the updated manuscript on lines 104–108 (main text) and lines 122–125 (figure legend).

Technical comments:

Comments 1: From healthy control (patients undergoing hip replacement), what was the anatomic site of samples collected and analyzed? These samples (S32-S39) showed only immune cells, no epithelial cells at all. Figure 2D also shows 12 healthy control samples (S32-43). Please show sub-labels for normal prostate tissue and bone tissues under the healthy control label.

Response 1: Thank you for your insightful comment. We have reviewed the metadata and confirmed that samples S32–S38 were derived from normal prostate tissue, while samples S39–S43 were collected from the proximal femur bone marrow of patients undergoing hip replacement surgery, representing healthy bone tissue. To clarify this distinction, we have added sub-labels under the “Healthy” group in Figure 2D to indicate “Prostate” and “Bone” samples. This update is reflected in the revised figure and its legend on line 159 of the manuscript.

Comments 2: While the study identifies UTY and USP9Y as critically involved in bone metastasis, the precise mechanisms underlying their function remain unclear. Could authors provide more clarity on these 2 Y chromosome localized genes? How are these involved in prostate cancer progression in the context of Androgen signaling, which plays an important role in prostate cancer progression? Please provide this discussion in the discussion section of the manuscript. 

Response 2: Thank you for your thoughtful comment. We agree that the roles of UTY and USP9Y in prostate cancer progression, particularly in relation to androgen signaling, require further clarification. In response, we have expanded the discussion section (lines 387–395) to provide additional details on the potential functions of UTY and USP9Y in modulating androgen receptor signaling and their relevance to the development of castration-resistant prostate cancer. We believe this addition adds important mechanistic context to our findings.

Comments 3: The study relies heavily on computational analyses and public datasets and lacks experimental validation of key findings, such as the role of UTY and USP9Y in bone metastasis. I understand that validating all the findings would be an unfair expectation here. However, I still suggest that the authors include a paragraph in the discussion to highlight actionable directions clearly.

Response 3: Thank you for this insightful suggestion. In response, we have added a new paragraph to the discussion section (lines 407–419) outlining specific experimental directions to validate our findings. These include Co-IP and ubiquitination assays to investigate the role of USP9Y in AR protein stabilization, in vitro functional studies under ADT conditions to assess the impact of UTY and USP9Y loss, as well as potential combination treatment strategies and epigenetic drug screening approaches. We believe this addition offers clear, actionable directions for future validation and therapeutic exploration.

Comments 4: Since this manuscript is heavy in bioinformatics analysis, I suggest that the authors add a glossary of terms for quick reference for the readers. A table of quick-referring reading text should explain the analysis in a few sentences, which will help readers with little or no bioinformatics background understand the analyses.

Response 4: Thank you for this valuable suggestion. We agree that including a glossary would assist readers—particularly those with limited bioinformatics experience—in understanding the analytical methods used in this study. In response, we have compiled a glossary of key terms and tools and provided it as Supplementary Table 1 for quick reference. This addition can be found on line 805 of the revised manuscript.

Comments 5: Also, I would strongly recommend expanding legends to provide more information rather than just one-liner for each subfigure. Currently, it is difficult to understand many of the subfigures based on the information provided in the legends.

Response 5: Thank you for your helpful suggestion. We agree that more detailed figure legends enhance clarity and help readers better interpret each subfigure. In response, we have revised and expanded the figure legends throughout the manuscript to include additional information such as the analytical methods used, the variables displayed, and relevant biological interpretations where appropriate. These updates can be found in the revised figure legends on lines 120–130, 160–168, 197–209, 265–277, 308–319, 340–349,737-748,750-762,764-773 and 791-803.

4. Response to Comments on the Quality of English Language

Point 1: No specific comments were raised regarding the quality of English language.

Response 1: We appreciate the reviewer’s positive evaluation. No further language revisions were required.

5. Additional clarifications

We have no additional clarifications at this stage. Thank you for your time and consideration.

Reviewer 2 Report

Comments and Suggestions for Authors

Feng et al. utilized publicly available single-cell datasets to systematically reveal the progression of bone metastasis in prostate cancer. This comprehensive study uncovers mechanisms involving mutation patterns, metabolic reprogramming, and microenvironment adaptation. Overall, it is an elegant study, providing useful information that could potentially illuminate new research directions.

Minor Issues:

  1. Check the authors’ names, especially “Yuan Sh.”
  2. The quality of the images is poor, especially Figure 1B. Please relabel all the panels with a larger font size.
  3. What is the color definition for Figure 1D?
  4. For neuroendocrine cells (NEndoCs), please use the abbreviation (NEndoCs) for the rest of the manuscript.
  5. Please label Figure 3C in a more detailed way.
  6. For Figure 3E, please label the X-axis (epithelial cell subpopulation (%)) or something appropriate.
  7. Please describe Figures 5C and 5D in the manuscript and ensure that all figures are described.
  8. For Figure 5C, it would be helpful to provide an explanation for why there are two groups of immune cells (ImmuCs) in the first level of the root.
  9. Since the NEndoCs subpopulation is a unique finding, please discuss it further in the “Discussion” section.

Author Response

Response to Reviewer X Comments

1. Summary

Thank you very much for taking the time to review our manuscript. We appreciate the reviewers’ thoughtful feedback and constructive comments, which have helped us improve the clarity and quality of our work. Please find below our detailed responses to each comment, and the corresponding revisions have been highlighted in yellow in the re-submitted manuscript. We have carefully addressed all concerns raised. Where we respectfully disagree, we have provided clear justifications and further clarification in our responses, and we welcome any additional guidance from the Academic Editor.

2. Point-by-point response to Comments and Suggestions for Authors

Comments 1: Check the authors’ names, especially “Yuan Sh.

Response 1: Thank you for your attention to the author details. We have carefully reviewed the author names and confirm that “Yuan Sh.” is correctly spelled and reflects the author's preferred name format. In accordance with the journal’s policy allowing only two corresponding authors, we have designated Yuan Sh. and Nianzeng Xing as the official corresponding authors. Xiuli Zhang, who was previously listed as a corresponding author, is now indicated as a co–first author to reflect her equal contribution to this work. These updates can be found on lines 5–6 of the revised manuscript.

Comments 2: The quality of the images is poor, especially Figure 1B. Please relabel all the panels with a larger font size.

Response 2: Thank you for your comment. We have replaced the low-resolution images and adjusted the font sizes across all figure panels, including Figure 1B, to improve visual clarity and consistency. The revised figures now feature higher resolution and enlarged, legible labels. These updates have been incorporated into the main manuscript and supplementary files, as reflected on line 119 of the revised version.

Comments 3: What is the color definition for Figure 1D?

Response 3: Thank you for pointing this out. In Figure 1D, each color represents an individual sample, illustrating inter-individual variability in the distribution of cell populations across the UMAP projection. To improve clarity, we have added a corresponding clarification to the figure legend. This update can be found on lines 126–128 of the revised manuscript.

Comments 4: For neuroendocrine cells (NEndoCs), please use the abbreviation (NEndoCs) for the rest of the manuscript.

Response 4: Thank you for your helpful suggestion. To improve clarity and ensure consistency throughout the manuscript, we have revised the text to uniformly use the abbreviation “NEndoCs” when referring to neuroendocrine cells. This standardization helps streamline the presentation, particularly in sections discussing multiple cell subtypes. The updated terminology appears on lines 39, 199, 213, 242, 319, 355, 362, 363, 367, 368, 371, 373, and 564 of the revised manuscript.

Comments 5: Please label Figure 3C in a more detailed way.

Response 5: Thank you for pointing this out. Figure 3C displays a heatmap of inferred copy number variations (CNVs) across epithelial cells, where each row corresponds to a single cell and each column represents a chromosomal region. To enhance clarity and interpretability, we have revised the panel label and expanded the figure legend to more clearly describe the content and relevance of this visualization. These updates can be found on lines 201–203 of the revised manuscript.

Comments 6: For Figure 3E, please label the X-axis (epithelial cell subpopulation (%)) or something appropriate.

Response 6: Thank you for your suggestion. To improve clarity, we have added an appropriate X-axis label to Figure 3E, indicating that each bar reflects the composition of epithelial cell subtypes within each sample. The label “Epithelial cell subtype (%)” has been included in the revised figure. This update can be found on line 196 of the manuscript.

Comments 7: Please describe Figures 5C and 5D in the manuscript and ensure that all figures are described.

Response 7: We appreciate your feedback and have revised the Results section to provide more detailed descriptions of Figures 5C and 5D. Figure 5C illustrates the log₂-transformed mutation burden across different epithelial subtypes, demonstrating that subtypes such as EMTCs and NEndoCs exhibit a higher mutational load, potentially reflecting more advanced or genetically unstable tumor states. Figure 5D depicts a pseudo-evolutionary trajectory inferred through the PAGA-tree method, indicating potential developmental relationships among epithelial subpopulations. These revisions are now included on lines 290–299 of the revised manuscript.     

Comments 8: For Figure 5C, it would be helpful to provide an explanation for why there are two groups of immune cells (ImmuCs) in the first level of the root.

Response 8: Thank you for this insightful observation. The presence of two ImmuC groups at the first level of the root in Figure 5D reflects transcriptionally distinct immune subclusters identified during trajectory analysis. These may correspond to functionally divergent immune cell populations—such as tumor-associated versus infiltrating immune cells—or represent different activation or differentiation states. We have added a clarifying note to the figure legend and included a brief explanation in the Results section on lines 294–299 of the revised manuscript.

Comments 9: Since the NEndoCs subpopulation is a unique finding, please discuss it further in the “Discussion” section.

Response 9: Thank you for highlighting this important point. We agree that the NEndoCs subpopulation represents a unique and potentially significant finding. In response, we have expanded the Discussion section (lines 362–373) to further elaborate on the distinct characteristics, enrichment patterns, and possible origins of NEndoCs. We also discuss their potential role in lineage plasticity and therapeutic resistance during prostate cancer progression, particularly in bone metastasis. We believe this addition strengthens the biological relevance and clinical implications of our findings.

3. Response to Comments on the Quality of English Language

Point 1: No specific comments were raised regarding the quality of English language.

Response 1: We appreciate the reviewer’s positive evaluation. No further language revisions were required.

4. Additional clarifications

We have no additional clarifications at this stage. Thank you for your time and consideration.
